# Vegetable Extracts as Therapeutic Agents: A Comprehensive Exploration of Anti-Allergic Effects

**DOI:** 10.3390/nu16050693

**Published:** 2024-02-29

**Authors:** Kazuhito Takemoto, Tian Ganlin, Masaki Iji, Takahiro Narukawa, Tomohisa Koyama, Luo Hao, Hiroyuki Watanabe

**Affiliations:** Graduate School of Human Life Sciences, University of Kochi Graduate School, 2751-1 Ike, Kochi 781-8515, Japan; tianganlin96@gmail.com (T.G.); masaki.iji02@gmail.com (M.I.); narukawa_takahiro@cc.u-kochi.ac.jp (T.N.); koyama_tomohisa@cc.u-kochi.ac.jp (T.K.); luohao612713@gmail.com (L.H.); watana@cc.u-kochi.ac.jp (H.W.)

**Keywords:** food allergies, polyphenols, RBL-2H3 cells, ovalbumin allergy, mast cell degranulation, Th1/Th2 balance, FcεRI subunits, anti-allergic effects

## Abstract

Food allergies are common worldwide and have become a major public health concern; more than 220 million people are estimated to suffer from food allergies worldwide. On the other hand, polyphenols, phenolic substances found in plants, have attracted attention for their health-promoting functions, including their anti-allergic effects. In this study, we examined the potential inhibitory effects of 80% ethanol extracts from 22 different vegetables on the degranulation process in RBL-2H3 cells. Our aim was to identify vegetables that could prevent and treat type I allergic diseases. We found strong inhibition of degranulation by extracts of perilla and chives. Furthermore, we verified the respective efficacy via animal experiments, which revealed that the anaphylactic symptoms caused by ovalbumin (OVA) load were alleviated in OVA allergy model mice that ingested vegetable extracts of perilla and chives. These phenomena were suggested to be caused by induction of suppression in the expression of subunits that constitute the high-affinity IgE receptor, particularly the α-chain of FcεR I. Notably, the anti-allergic effects of vegetables that can be consumed daily are expected to result in the discovery of new anti-immediate allergenic drugs based on the components of these vegetables.

## 1. Introduction

Food allergies (FAs) are very common worldwide and have become a major public health concern. Although precise epidemiological data are said to be lacking [1], according to Martinis et al., the prevalence of food allergies has increased considerably in Western countries over the past two decades, reaching 10% in preschool children [2]. It is estimated that more than 220 million people suffer from FAs worldwide [1].

On the other hand, the prevalence of FAs varies geographically due to cultural influences on eating habits; according to Gargano [3], peanut allergy is one of the most common causes of food anaphylaxis in the United States, the United Kingdom, and Australia but is rare in Italy and Spain, where peanut consumption is low. Peach allergy is more frequent in southern Europe, where peach consumption is high, and is very rare in northern Europe [4]. In addition, anaphylactic reactions have been reported after ingestion of carmine red (E120), which is obtained from insects such as beetles (mealworms, superworms, and silkworms) and cochineal beetles such as *Dactylopius coccus* and is used as a coloring agent in the food industry [5,6,7,8]. With the globalization of dietary habits, the number of food ingredients that cause allergic reactions is expected to increase worldwide.

Nevertheless, there have been few reports showing anti-allergic effects of daily vegetable ingredients; Shimoda et al. evaluated the anti-type I allergic effects of aqueous ethanol extract from the above-ground part of Japanese butterbur, which is used as a domestic vegetable in Japan, using rats and RBL-2H3 mast cells [9].

Polyphenols are a large group of phytochemicals containing phenolic rings with two or more hydroxyl groups. Typically, a single plant contains multiple polyphenols, which constitute components of its pigments, bitterness, and astringency [10]. Polyphenols have attracted attention for their many health-promoting functions and have been reported to possess antioxidant [11], anti-allergic [12], anti-obesity [13], osteoporosis prevention [14], and cognitive function maintenance effects [15]. Furthermore, quercetin in vegetables has been reported to have negative effects on FcεRI cross-linking and other activated receptors in mast cells and exhibits anti-allergic properties [16].

Accordingly, scholars should focus on the fact that polyphenols are present in most vegetables and can be consumed daily, thereby exhibiting potential anti-allergic effects. This study aimed to examine the possible anti-allergic effects of vegetable extracts. Therefore, the purpose of this study was to identify vegetables for the prevention and treatment of type I allergic diseases and to verify their effectiveness through animal experiments. Extracts from 22 vegetable species were prepared using ethanol, and their ability to inhibit the degranulation of RBL-2H3 cells was evaluated. Furthermore, vegetable extracts with strong inhibitory effects on cell degranulation were evaluated for their ability to suppress allergic sensitization in mice.

## 2. Materials and Methods

### 2.1. Sample Preparation

Twenty-two types of vegetables were used: tomatoes, green peppers, shishito peppers, pumpkins, snap peas, broccoli, komatsuna, spinach, crown daisies, green onions, perilla, chives, asparagus, carrots, eggplant, bitter gourds, cabbage, onions, garlic, myoga, radish, and ginger. All vegetables were purchased from JA Kochi (Kochi, Japan). Moisture content of the perilla and chives was measured using a simple moisture meter (A&D Company, Ltd.; Tokyo, Japan), and 99.9% ethanol was added to bring the ethanol concentration to 80%. The mixture was then homogenized using a Waring multi-blender (CB-15T, FMI Corporation, Tokyo, Japan) and a Polytron homogenizer (IKA T18, IKA^®^ Japan K.K., Osaka, Japan), filtered, and centrifuged at 1610× *g* for 10 min. After filtration, the supernatant was centrifuged at 1610× *g* for 10 min, and the supernatant was separated. The supernatant liquid was eliminated via a rotary evaporator to eliminate the solvent, then subjected to drying using a lyophilizer, resulting in the formation of a powdered substance.

### 2.2. Determination of Total Polyphenols

Total polyphenol content was determined using the Faurin–Ciocarto method [17] (converted to chlorogenic acid content). Briefly, 80% ethanol extracts of all vegetables were dissolved in 5% metaphosphoric acid solution, 0.4 M sodium carbonate solution, and phenol reagent. After 30 min at room temperature, the absorbance was measured at 765 nm.

### 2.3. Determination of Antioxidant Capacity

Antioxidant capacity was measured using the 1,1-diphenyl-2-picrylhydrazyl (DPPH) method [18] (converted to the amount of L-ascorbic acid). Using 80% ethanol extracts of each vegetable, DPPH methanol solution was added after dissolving in 100 mM Tris hydrochloric acid buffer at pH 7.4 and left at room temperature for 20 min. Subsequently, the absorbance was measured at 520 nm.

### 2.4. Laccase Treatment of Vegetable Extracts

Laccase (Daiwa Fine Chemicals Co., Ltd., Hyogo, Japan) was dissolved in 0.01 M sodium acetate buffer (pH 4.5) to make 10 mg/mL of enzyme solution. After 24 h, the reaction was terminated by heating the solution in boiling water for 10 min.

### 2.5. Cell Model

Rat basophilic leukemia cells (RBL-2H3), provided by the RIKEN BioResource Center (Ibaraki, Japan), were used as the cell model. Cell cultures were grown in low glucose Dulbecco’s modified Eagle’s medium (DMEM, Nacalai Tesque, Inc., Kyoto, Japan) containing antibiotics (Nacalai Tesque, Inc., Kyoto, Japan).

### 2.6. Evaluation of Cell Viability of Vegetable Extracts on RBL-2H3 Cells

RBL-2H3 cells (2.0 × 10^5^ cells/well) were seeded in 96-well plates and cultured for 48 h. Subsequently, 80% ethanol extracts of each vegetable, with a concentration gradient of ≤1000 µg/mL, were added, and the cells were incubated for another 20 h. The extracts were dissolved and diluted in DMSO (at a concentration of 0.5%) in the medium. Ten microliters of Cell Counting Kit-8 (Dojindo Molecular Technologies, Inc., Kumamoto, Japan) was added and allowed to react for 4 h. After 4 h, the absorbance was measured at 450 nm using a microplate reader.

### 2.7. Degranulation Test via Hexosaminidase Release Examination

Cells (2.0 × 10^5^ cells/well) were seeded in 96-well microplates and cultured overnight. The medium was then removed and replaced with 100 µL medium containing 50 ng/mL anti-DNP-IgE antibody (Yamasa corporation, Chiba, Japan) and cultured for 18 h. The cells were then washed once with MT buffer. One hundred microliters of each vegetable extract solution was added and allowed to preincubate for 10 min. Each vegetable extract was dissolved in MT buffer at a concentration that did not affect the cell viability. Then, 10 µL of DNP-HSA (25 µg/mL; Sigma-Aldrich Japan, Tokyo, Japan) was added to each well as an antigen and incubated for another 2 h. The same reaction in the MT buffer without the sample was used as a control. After 10 min of ice-cooling, 50 µL of the culture supernatant was transferred to a 96-well plate, and 0.1% Triton X-100/MT buffer was added to the cells. Intracellular fluid was obtained by cell disruption with an ultrasonic device. Fifty microliters of the intracellular fluid was transferred to a 96-well plate, warmed at 37 °C for 5 min, and then dissolved in 100 µL of 0.1 mol/L citrate buffer (pH 4.5) with 3.3 mmol/L p-nitrophenyl-2-acetoamido-2-deoxy-β-D glucopyranoside (FUJIFILM Wako Pure Chemical Corporation, Osaka, Japan). Subsequently, β-hexosaminidase was allowed to react with the fluid at 37 °C for 25 min. The reaction was stopped by adding 100 µL of 2.0 mol/L glycine buffer (pH 10.4) to the reaction solution, and the absorbance at 405 nm was measured using a plate reader. The laccase-treated extracts of perilla and chives were also used as experimental samples, and the β-hexosaminidase release activity was measured.

### 2.8. Intracellular Calcium Ion Evaluation

Intracellular calcium ions were measured using Calcium Kit-Fluo 4 (Dojindo Molecular Technologies, Inc., Kumamoto, Japan) as follows: 2.0 × 10^5^ cells/well were seeded in 96-well black microplates and cultured as per the degranulation test described prior to sensitize the cells to IgE antibodies. Cells were then washed once with PBS, and 100 µL/well of loading buffer adjusted with the reagent supplied with the kit was added and incubated for 1 h. The cells were then washed once with PBS, and 100 µL of perilla or chive extract solution was added and allowed to preincubate for 10 min. Each vegetable extract was processed as described above and dispersed in PBS. Subsequently, each vegetable extract solution was removed and replaced with 100 µL of recording medium prepared with the kit’s supplied reagent. Ten microliters of DNP-HSA (25 µg/mL; Sigma-Aldrich Japan, Tokyo, Japan) was added to each well as an antigen, and fluorescence intensity was immediately measured at λex = 485 nm and λem = 538 nm. The control group lacked any vegetable extract, while the blank group was deemed insensitive to IgE.

### 2.9. Animals and Experimental Diets

This study was approved by the Animal Experiment Committee of Kochi Prefectural University (Approval No. 2022-7). Animal experiments were conducted in accordance with the animal experiment regulations of Kochi Prefectural University. The animal laboratory was maintained at a room temperature of 24 ± 2 °C, humidity of 55 ± 3%, and a 12-h light/dark cycle (light period: 7:00 AM to 7:00 PM, dark period: 7:00 PM to 7:00 AM). Mice introduced from the breeders were fed an MF diet (Oriental Yeast Co., Ltd., Tokyo, Japan) and drinking water (tap water) ad libitum and were pre-bred for 7 days. The experimental animals comprised 4-week-old female ICR (Slc: ICR) mice (Japan SLC, Shizuoka, Japan).

Feed ingredients included casein (FEED ONE Co., Ltd. Kanagawa, Japan), l-cysteine (FUJIFILM Wako Pure Chemical Corporation), cornstarch (Marusan, Kochi, Japan), edible oil (vitamin E-free) (Tama Biochemical Co., Ltd., Tokyo, Japan), cellulose powder (Oriental Yeast Co., Ltd., Tokyo, Japan), AIN-93G mineral mixture (Oriental Yeast Co., Ltd., Tokyo, Japan), AIN-93 vitamin with choline deuterite (Oriental Yeast Co., Ltd.), and tertiary butylhydroquinone (FUJIFILM Wako Pure Chemical Corporation, Osaka, Japan).

### 2.10. Study Design

After preliminary rearing, the mice were divided into four groups to ensure that their mean body weights were approximately equal: nonallergic (NA) (*n* = 9), allergic (A) (*n* = 8), perilla (P) (*n* = 8), and chives (C) (*n* = 8). The rearing diet was based on AIN 93G (normal diet), while the O group was fed a perilla diet with perilla extract, and the N group was fed a chive diet with chive extract added to the diet (Table 1).

The feed was administered using the pair-feeding method so that the daily caloric intake of each group was the same, and water was provided ad libitum. From the 1st day of rearing to the 21st day, all groups were fed freely on a normal diet; from the 21st day of rearing, groups NA and A were fed an AIN93 normal diet, group P was fed a large-leaf diet, and group C was fed a chive diet.

Group A, P, and C mice were sensitized to OVA by intraperitoneal administration of a mixture of aluminum hydroxide adjuvant and OVA on days 7 and 21 of rearing; the OVA mixture was prepared by mixing 1 mg/mL OVA solution and aluminum hydroxide adjuvant in a 3:1 proportion; for the NA group, 200 µL/animal was administered intraperitoneally of a 3:1 proportionate mixture of PBS and aluminum hydroxide adjuvant.

On the 40th day of rearing, feces were collected for 24 h. The feces were freeze-dried and stored at 4 °C until further use. On the last day of rearing, OVA solution was orally administered to the A, P, and C groups at a dose of 6 mg/animal to induce anaphylaxis. In the evaluation of anaphylactic shock, rectal temperature and auricular thickness were measured, and shock symptoms were observed. Rectal temperature was measured before and 5 min after oral administration of the antigen using a temperature-measuring device (AD-1687, A&D Company, Ltd., Tokyo, Japan) connected to a temperature-measuring probe (AX-KO4746-100, A&D Company, Ltd.). The temperatures were measured 5, 15, and 30 min after oral administration of the antigen, and the temperature changes were compared. Allergic symptoms in mice 30 min after oral antigen administration were also observed and scored on a 5-point scale (Table 2), as described by Li et al. [19]. In addition, mice were immediately sacrificed by cardiac blood sampling under isoflurane inhalation anesthesia. The spleen, large intestine, and cecum were collected. Each organ was stored at −30 °C until further analysis.

### 2.11. Preparation of Fecal Samples

Five hundred microliters of PBS was added to 50 mg lyophilized feces and homogenized with a homogenizer (T10 basic, Yamato Scientific Co., Ltd., Tokyo Japan). The resulting fecal suspension was used for the measurement of IgA in feces.

### 2.12. Determination of Antibody Concentrations in Blood and Feces

#### 2.12.1. Fecal IgA

Anti-mouse IgA antibody (Rabbit-poly, Funakoshi Co., Ltd., Tokyo, Japan) was diluted to 100 ng/mL in PBS as a primary antibody, seeded at 100 µL/well in a 96-well plate, and allowed to adsorb overnight at 37 °C. After an overnight incubation, the primary antibodies were removed. After 1 h, the blocking solution was removed and washed with PBS-T, and 100 µL/well of standard antibody and fecal suspension diluted 200-fold was seeded at 100 µL/well and allowed to react at 37 °C for 2 h. Subsequently, the solution was removed from the wells, and the cells were washed with PBS-T. As a secondary antibody, anti-mouse IgA HRP-conjugated antibody (Goat-poly, Novus Biologicals, Centennial, CO, USA) was diluted to 100 ng/mL in PBS and seeded at 100 µL/well. After 2 h, the solution was removed from the wells and the cells were washed with PBS-T. After washing, 100 µL/well of chromogenic reagent was seeded and allowed to develop at 37 °C for 25 min. After this, 100 µL of reaction stopper solution was added to stop the reaction, and absorbance was measured at 492 nm.

#### 2.12.2. Total IgE in the Blood

Anti-mouse IgE antibody (Rabbit-poly, Funakoshi Co., Ltd., Tokyo, Japan) was diluted to 100 ng/mL in PBS as a primary antibody, seeded at 100 µL/well in a 96-well plate, and allowed to adsorb at 37 °C overnight. The primary antibody was then removed and washed with PBS-T, and 100 µL/well of blocking solution was seeded and allowed to adsorb for 1 h. The blocking solution was then removed and washed with PBS-T, and 100 µL/well of standard antibody and 200-fold diluted sample solution were seeded in the wells and allowed to react at 37 °C for 2 h. The solution was removed from the wells, and the cells were washed with PBS-T. As a secondary antibody, anti-mouse IgA HRP-conjugated antibody (Goat-poly, Novus Biologicals, Centennial, CO, USA) was diluted to 100 ng/mL in PBS, seeded at 100 µL/well, and reacted for 2 h at 37 °C. The antibody was then diluted to 100 ng/mL in PBS and seeded at 100 µL/well. The reaction was performed for 2 h at 37 °C. The solution was then removed from the wells, and the cells were washed with PBS-T. The cells were then seeded with 100 µL/well of chromogenic reagent and incubated at 37 °C for 25 min. The reaction was stopped by adding 100 µL of reaction stopper solution, and absorbance was measured at 492 nm.

#### 2.12.3. Measurement of OVA-Specific IgE Antibody Titers in the Blood

Serum was used as the sample, and OVA was prepared at 100 µg/mL, seeded at 100 µL/well in a 96-well plate, and allowed to adsorb at 37 °C overnight. The OVA solution was then removed and washed with PBS-T, and 100 µL/well of blocking solution was seeded and allowed to adsorb for 1 h. The blocking solution was then removed and washed with PBS-T, and 100 µL/well of standard antibody and 200-fold diluted serum sample solution were seeded in the wells and allowed to react at 37 °C for 2 h. Subsequently, the solution was removed from the wells, and the cells were washed with PBS-T. Then, anti-mouse IgE HRP-conjugated antibody (Rat-mono, abcam plc, Cambridge, UK) was diluted to 100 ng/mL in PBS, seeded at 100 µL/well, and reacted for 2 h at 37 °C. The solution was removed from the wells and washed with PBS-T, and 100 µL/well of chromogenic reagent was seeded. The wells were then washed with PBS-T and incubated at 37 °C for 25 min. The reaction was stopped by adding 100 µL of reaction stopper solution, and absorbance was measured at 492 nm.

### 2.13. RT-PCR

RNA solutions were prepared using Agencourt RNAdvance Tissue (Beckman Coulter Inc., Brea, CA, USA) and stored at −80 °C until time of use. From the RNA solution, cDNA was prepared by reverse transcription using a PrimeScript™ RT reagent kit (Takara Bio Inc., Shiga, Japan). cDNA solution was obtained by reverse transcription reaction at 37 °C for 15 min, followed by treatment at 85 °C for 5 s to inactivate the reverse transcriptase.

Real-time PCR was performed using the Probe qPCR Mix (Takara Bio Inc., Shiga, Japan). The expression levels of TBX21; GATA3; Foxp3; Rorc; and FcεRⅠα, β, and γ genes in the spleen and colon tissues were quantified using TaqMan^®^ probes (Thermo Fisher Scientific, Tokyo, Japan) as monitoring reagents. Probes for each gene were purchased from Thermo Fisher Scientific (Tokyo, Japan). The obtained results were corrected with actin (Thermo Fisher Scientific K.K. Tokyo, Japan: Mm00607939_s1) as an internal standard, and the relative gene expression levels of each group were calculated relative to the gene expression levels in the non-sensitized group. The TBX21/GATA3 expression ratio was calculated as an index of the Th1/Th2 balance.

### 2.14. Statistical Analysis

The obtained values were expressed as mean ± standard deviation for each group. The statistical software BellCurve for Excel Ver. 4.06 (Social Survey Research Information Co., Ltd., Tokyo, Japan) was used for statistical analysis.

A *t*-test was used to compare vegetable extracts with and without laccase treatment, and Dunnett’s test was used to compare intracellular calcium concentrations using the control after analysis of variance. Otherwise, the Bonferroni weight comparison test was used. The significance level was set at less than 0.05 or 0.01.

## 3. Results

### 3.1. Total Polyphenol Content and Antioxidant Capacity of the 22 Vegetable Extracts

Table 3 shows the total polyphenol content and antioxidant capacity of the 22 vegetables’ 80% ethanol extracts. The total polyphenol content of each vegetable’s 80% ethanol extract was converted to the relative amount of chlorogenic acid in 1 g of the dried extract, and the vegetable extract with the highest total polyphenol content was perilla (115.94 ± 7.84 mg/g). The highest antioxidant capacity of the 80% ethanol extract of each vegetable as shown by the DPPH method was observed in myoga (1.92 ± 0.01 mg/g).

### 3.2. Cell Viability and Degranulation Inhibition Activity of the 22 Vegetable Extracts

Table 4 shows the concentrations of each vegetable’s 80% ethanol extract added that did not affect cell viability and the maximum β-hexosaminidase inhibition rate and concentration of each vegetable extract added without affecting cell viability.

Vegetables with significant inhibition of release compared with those without addition included garland chrysanthemum 26.62 ± 4.43%, perilla 44.16 ± 4.34%, chives 49.18 ± 3.51%, and myoga 18.88 ± 2.63%. Of these vegetables, chives and perilla were the only ones that showed concentration-dependent inhibition of release, whereas the inhibition by crown daisy and myoga was not concentration-dependent. No inhibition of release was observed for the other vegetables. The IC50 values of the perilla and chive extracts, which showed concentration-dependent degranulation inhibition, were 602.5 µg/mL and 125.4 µg/mL, respectively.

Figure 1 shows the results of the comparison of perilla and chive extracts with and without laccase treatment. The degranulation inhibitory activities of both perilla and chive extracts were markedly reduced by laccase treatment at all concentrations.

### 3.3. Intracellular Calcium Concentration Changes during Degranulation Reaction

Figure 2 shows the changes in intracellular calcium concentration during the degranulation test. In the degranulation reaction without vegetable extract shown in the figure, an increase in the intracellular calcium concentration was observed immediately after the addition of the antigen. In contrast, no increase in the intracellular calcium concentration was observed with the addition of perilla and chive extracts after the addition of the antigen.

### 3.4. Weight Change in Mice during the Rearing Period

The body weights of the mice during the 6-week period are shown in Figure 3. No differences in body weight were observed between the groups during any weight measurement.

### 3.5. Anaphylactic Symptoms

The subjective score of allergic symptoms 30 min after oral administration of OVA is shown in Figure 4A, and the rectal temperature 30 min after OVA administration is shown in Figure 4B.

The anaphylactic symptom subjective score of the NA group showed no symptomatic individuals; the score of the A group was 2.88 ± 0.35, which was considerably higher than that of the NA group; the anaphylactic symptom subjective scores of the P and C groups were remarkably lower than that of the A group. However, no differences were observed in terms of rectal temperature between the P and C groups. At 30 min after OVA administration, the temperature was higher in groups P and C compared with group A, and the difference between groups A and C was found to be remarkable.

### 3.6. Analysis of Feces

Figure 5A shows the 24 h fecal weights. Figure 5B demonstrates that the amount of IgA in the feces was considerably lower in group A than in the other three groups. Figure 5D represents the cecal pH. The cecal pH was markedly higher in group N than in group A.

### 3.7. Blood Antibody Titer

Figure 6A shows that the total IgE in the blood of the other groups was markedly higher than that in the NA group. Compared with group A, the total IgE in groups P and C was notably lower.

Figure 6B shows the concentration of OVA-specific IgE in the blood; no OVA-specific IgE was detected in the blood of the NA group. Figure 6C shows the OVA-specific IgE/total IgE ratio, which was markedly lower in groups P and C than in group A.

### 3.8. Gene Expression in the Colon

Figure 7A shows that the intensity of TBX21 expression in the colon was considerably lower in group A than in group NA. Furthermore, it was markedly higher in groups P and C than in group A. Figure 7B shows that the expression intensity of GATA3 in the colon was markedly higher in group A than in group NA. In contrast, the expression of GATA3 in groups P and C was considerably lower than that in group A. Figure 7C shows the expression intensity of Foxp3 in the colon, which was markedly higher in the P and C groups than in the NA and A groups. Figure 7D shows Rorc expression intensity in the colon, which was markedly lower in groups P and C than in groups NA and A. Figure 7E shows the gene expression intensity ratio of TBX21 and GATA3 in the colon.

Figure 8 shows the gene expression intensities of FcεR Iα, β, and γ in the colon; the gene expression level of FcεR Iα was extensively decreased in groups NA, P, and C compared with that in group A, a decrease that was more pronounced in groups P and C. The gene expression level of FcεR Iβ was markedly decreased in groups NA, P, and C compared with that in group A. The gene expression level of FcεR Iβ was considerably decreased in groups P and C compared with that in group A. The gene expression level of FcεR Iγ was notably decreased in groups NA, P, and C compared with that in group A. The gene expression level of FcεR Iγ was markedly decreased in groups NA, P, and C compared with that in group A. However, gene expression of FcεR Iβ and FcεR Iγ was not as significantly suppressed as that of FcεR Iα.

### 3.9. Gene Expression in the Spleen

Figure 9A shows TBX21 expression intensity in the spleen. The intensity of TBX21 expression in the spleens of group A was markedly lower than that in group NA and considerably higher in groups P and C than in group A. Figure 9B shows the GATA3 gene expression intensity in the spleen, which was markedly higher in group A than in group NA. In contrast, the expression intensity of GATA3 in the spleens of groups P and C was notably lower than that in group A. Figure 9C shows the Foxp3 expression intensity in the spleen, which was markedly higher in groups P and C than in groups NA and A. Figure 9D shows the intensity of Rorc expression in the spleen, which was considerably higher in group A than in group NA. In contrast, that of group C was markedly lower than that of group A. Figure 9E shows the gene expression intensity ratio of TBX21 and GATA3 in the spleen; the gene expression intensity ratio of TBX21 and GATA3 in the spleens of group A was lower than that of the other three groups.

Figure 10 shows the gene expression intensities of FcεR Iα, β, and γ in the colon. Gene expression of FcεR Iα was markedly lower in groups NA, P, and C compared with that in group A. Gene expression of FcεR Iβ was lower in groups NA, P, and C compared with that in group A. Gene expression of FcεR Iγ was considerably decreased in groups NA, P, and C compared with that in group A.

## 4. Discussion

In this study, we found remarkable inhibitory effects on degranulation in RBL-2H3 cells when exposed to ethanol extracts from 22 vegetable species commonly used in Japan, including garland chrysanthemum, perilla, chives, and myoga. In particular, perilla and chives maintained cell viability even when added to cells at high concentrations and had the highest maximum degranulation inhibitory effect. In addition, a comparison between chives (IC50: 125.4 µg/mL) and perilla (IC50: 602.5 µg/mL) revealed that both the degranulation inhibition rate and its IC50 were superior in the case of chives.

The amount of extract added, which showed the maximum value of degranulation inhibition by the ethanol extracts of the 22 vegetables, did not show any correlation with total polyphenol content or antioxidant capacity. However, compared with the total polyphenol content in chives, the total polyphenol content in leaves was more than 12-fold higher, suggesting that chives contain polyphenols that inhibit degranulation in smaller amounts compared with leaves. In other words, for the polyphenols present in the extracts, the difference in type rather than quantity suggests that they are strongly involved in the inhibition of degranulation.

Laccase catalyzes various oxidation reactions, including the degradation and polymerization of polyphenols [20]. Treatment of chives and perilla with laccase ethanol extracts abolished the inhibitory effect on degranulation. This suggests that the inhibitory effect of chives and perilla on degranulation is polyphenol-dependent. It was also inferred that the degranulation inhibitory effect per unit weight of polyphenols was stronger in chives than in perilla. Chives are known to contain kaempferol as a representative polyphenol [21], while luteolin and rosmarinic acid have been reported as polyphenols in perilla [22], suggesting that these polyphenols are involved in degranulation inhibition.

In IgE-dependent anaphylaxis, mast cell activation is believed to increase vasodilation and vascular permeability, accompanied by a decrease in blood pressure and body temperature [23]. Therefore, suppression of the decrease in body temperature after antigen ingestion is an indicator of relief from anaphylactic symptoms. Thirty minutes after antigen administration, the decrease in body temperature was suppressed by the intake of perilla and chive extracts. Particularly, in the C group, a distinct difference was evident compared with the group that did not receive the vegetable extracts. This suggests that the intake of chive extract strongly suppressed anaphylaxis. In the subjective evaluation of anaphylaxis, allergic symptom scores were reduced in the P and C groups, with the C group showing stronger suppression of anaphylactic symptoms, consistent with the evaluation using RBL-2H3 cells, suggesting that chives have strong anti-allergic properties.

Inhibition of antigen entry is postulated to be one of the mechanisms by which anaphylaxis is suppressed. In food allergies, entry and contact with antigens occur through the gastrointestinal tract, and vegetable extracts may affect the defense mechanisms of the gastrointestinal tract.

The surface of the intestinal mucosa forms a physical and chemical barrier, including tight junctions between epithelial cells, the mucin layer, antimicrobial peptides, and secretory IgA antibodies, which prevent foreign substances such as bacteria from entering the body [24].

In addition to the intrinsic mucosal layer, the intestinal bacteria are involved in immunity [25]. An increase in intestinal bacteria increases the cecum weight and decreases its pH owing to an increase in short-chain fatty acid production. In contrast, no difference in cecal weight was observed among the groups in this study. No decrease in intracecal pH was observed with vegetable extract intake. Thus, in the present study, the anti-allergic effects of perilla and chive extracts were presumed to be due to a mechanism other than the involvement of intestinal bacteria.

IgA, which plays a central role in the intestinal immune system, is secreted into the intestinal tract, blocks the entry of pathogens and foreign substances, and acts as a neutralizing antibody [26]. Furthermore, antigen-specific IgA forms complexes with antigens and sterically blocks epitopes to suppress IgE-dependent anaphylaxis [27]. Furthermore, IgA modulates IgE-induced activation of basophils and mast cells [28] and plays various roles in alleviating food allergy symptoms.

In the vegetable extract-supplemented diet group, the amount of IgA in the feces was markedly higher than that in group A, suggesting that IgA may be involved in the alleviation of allergic symptoms, including the inhibition of allergen entry. Since there was no difference in fecal IgA levels between the NA and the P or C groups, it can be inferred that this was the result of a correction of IgA secretion that was disrupted by antigen sensitization due to vegetable intake.

In this study, the activation of Th1, Th2, Th17, and Treg cells was evaluated using the gene expression levels of TBX21, GATA3, Rorc, and Foxp3 [29] transcription factors of cell proliferation, respectively.

Naive T cells differentiate into effector T cells (Th1, Th2, and Th17) depending on the type of cytokines produced by the invading pathogen [30]. These cells are normally balanced and are responsible for the immune response; however, allergic diseases are thought to develop when the balance of these functions is disrupted and Th2 cells become dominant [31].

The expression level of the TBX21 gene was markedly elevated in the P and C groups compared with that in the A group, whereas the expression level of the GATA3 gene, a Th2 transcription factor, was markedly lower in the P and C groups than in the A group. This implies that the Th cell balance, initially skewed towards Th2 cells due to allergy induction, was corrected by the consumption of perilla and chive extracts, leading to a marked shift towards Th1 dominance and improvement in the allergic state.

Th17 cells induce autoimmunity and inflammation, whereas Tregs suppress these events to maintain immune homeostasis. Therefore, elucidating the mechanisms that influence the Th17/Treg cell balance is crucial for a deeper understanding of autoimmunity and tolerance [32].

In this study, Rorc gene expression, a transcription factor for Th17 cell proliferation, was markedly decreased in both sensitized groups, P and C, compared with that in the A group. This suggests that Th17 cell proliferation was suppressed by intake of perilla and chive extracts.

Foxp3 gene expression, a transcription factor of Treg cells, was markedly upregulated in the sensitized groups, P and C, compared with that in group A, suggesting that it contributes to the maintenance of immune system homeostasis by Tregs. Thus, the results suggest that perilla and chive extracts have an ameliorative effect on the balance of Th17/Treg cells in the large intestine.

The spleen, like the intestinal tract, plays an important role in immune system function. Our results suggest that in the spleen, as in the colon, Th cell transcription factors correct the Th1/Th2 balance disrupted by antigen sensitization and mitigate the induction of Th17 cell differentiation that contributes to allergen induction. This suggests that the effects of compounds derived from the extracts of perilla and chives extend not only to the large intestine, where each component is in direct contact, but also to the whole body via the blood.

Total IgE antibodies in the blood were low in the non-sensitized group but elevated in the sensitized group. This result was characteristic of food allergic reactions after the sensitized group was administered OVA as an allergen. In contrast, the total IgE concentration in the blood of the P and C groups was lower than that in group A, suggesting that the allergic reaction was reduced by vegetable extract intake. This is also evident from the blood IgE results specific to OVA.

In type I allergic reactions, the activation of mast cells and basophils is induced by allergen-induced cross-linking and aggregation of a threshold number of allergen-specific IgE-charged FcεRI on their surface [33]. In groups P and C, the ratio of OVA-specific IgE to total IgE was also reduced, and FcεRI allergen-induced cross-linking and aggregation may also be reduced in these groups.

FcεRI, discussed earlier, is an IgE (Fc fragment of IgE antibody)-specific receptor with high binding affinity [34,35]. Human FcεRI is expressed primarily on mast cells and basophils. The FcεRI complex on the surface of human mast cells is a tetramer consisting of one α-subunit, one β-subunit, and two γ-subunits (αβγ2) [34,35]. Of these, the α-subunit contains two extracellular Ig superfamily domains (α1 and α2), both of which are responsible for ligand (Fc fragment of IgE antibody) binding [36]. On the other hand, gene transfer experiments have shown that human FcεRI is expressed on the cell surface not only in the αβγ2 tetrameric structure but also in the αγ2 trimeric structure [37]. In addition, the β- and γ-subunits have an immunoreceptor tyrosine-based activation motif (ITAM) along with a Sarcoma (SRC) homology 2 (SH2) domain, providing binding sites for various intracellular signaling molecules [35]. In rodents, the structure of FcεRI is tetrameric, consisting of α-, β-, and γ-chain dimers associated by non-covalent bonds. In mice, the β chain is essential for FcεRI to be expressed on the cell surface, and only the αβγ2 tetrameric structure exists [35].

Type I allergies are triggered by antigen entry. The invading antigen cross-links IgE antibodies on the surface of basophils and initiates signal transduction, resulting in the release of allergy-causing substances such as via degranulation. Notably, after IgE binding to FcεRI, signaling is followed by protein kinase phosphorylation of the substrate protein and activation or inactivation of the protein, which then transmits the signal. Signal transduction can be categorized into Ca^2+^-dependent and Ca^2+^-independent pathways. The Lyn-Syk-LAT and Lyn-Btk-PLCγ pathways fall under the Ca^2+^-dependent pathways, while the Fyn-Gab2-PI3K pathway is classified as Ca^2+^-independent [38,39].

Since all IgE-mediated responses involve signaling through FcεRI, targeting FcεRI, or its pathway components, is considered an ideal strategy to prevent food-induced responses [40]. These methods include (1) preventing IgE binding to FcϵRI, (2) reducing B cell production of IgE, and (3) inhibiting FcϵRI signaling. These methods have been used to develop anti-allergic agents.

In recent years, based on the inhibition of FcϵRI activation, naturally occurring anti-allergic substances that inhibit allergic reactions have been reported, such as tannins, catechins, peptides, and oligosaccharides [41]. Apigenin and luteolin in sesame and polyphenol extracts from the pomace of black and white sesame have also been reported to decrease the phosphorylation of Syk, PLCγ2, and Akt [42]. Resveratrol inhibits splenic tyrosine kinase (Syk) and phospholipase Cγ (PLCγ) in mast cells [43].

Using human basophil KU812F cells, Shim et al. have shown that 6-methoxyl luteolin isolated from *Chrysanthemum zawadskii* potently inhibits histamine release and calcium influx via downregulation of the FcεRI α-chain [44]. Tamura et al. reported that 2’,3’-dihydroxypberlin from the South American medicinal plant *Verbascum thapsus* L. downregulates the mRNA level of the β chain of FcεRI [45]. For the γ chain subunit, 7- O-methylglycitein and tectrigenin extracted from the medicinal plant Puerariae Flos have been reported to suppress gene expression [46].

In the present study, administration of ethanol extracts of chives and perilla to OVA-sensitized mouse models suppressed gene expression of all subunit proteins constituting FcεRI compared with controls. The results of this study show that treatment of the sensitized group with extracts of perilla and chives markedly suppressed gene expression of the α subunit constituting FcεRI in the colon and considerably suppressed the β subunit. In the spleen, gene expression of the α and γ subunits was suppressed. The key mechanism underlying the anti-allergic effect of perilla and chives may involve the suppression of gene expression of the subunits constituting FcεRI.

One limitation of this study is that it was validated in an in vitro cell model using RBL-2H3 and in an animal model using ICR mice, which may not fully reproduce human responses. As already mentioned, it is possible that the alleviation of allergic symptoms via the effect on FceR Iβ expression may not be fully demonstrated in humans, because the FceR I expressed on the cell surface is only αβγ2 type in mice, while αγ2 type is also present in humans.

## 5. Conclusions

In conclusion, 80% ethanol extract of perilla and chives inhibited degranulation of RBL-2H3 cells. Perilla and chive extracts inhibited anaphylaxis after oral OVA administration in OVA-sensitized ICR mice. The mechanism was suggested to be related to the suppression of OVA-specific IgE production and the suppression of gene expression of the α and β subunits of FcεRI in the colon and the α and γ subunits of FcεRI in the spleen.

The results of this study are expected to contribute to the development of new anti-allergy drugs based on the anti-allergic effects of vegetables that can be consumed daily and the components contained in these vegetables.

## Figures and Tables

**Figure 1 nutrients-16-00693-f001:**
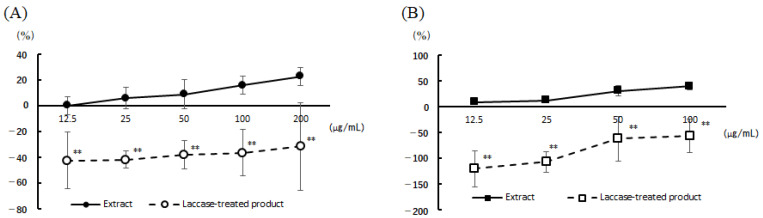
Inhibition of β-hexosaminidase release by laccase treatment. (**A**) Perilla extract, (**B**) chive extract. Values are mean ± standard deviation (*n* = 4). Significant differences at the same concentration were tested by *t*-test. ** *p* < 0.01.

**Figure 2 nutrients-16-00693-f002:**
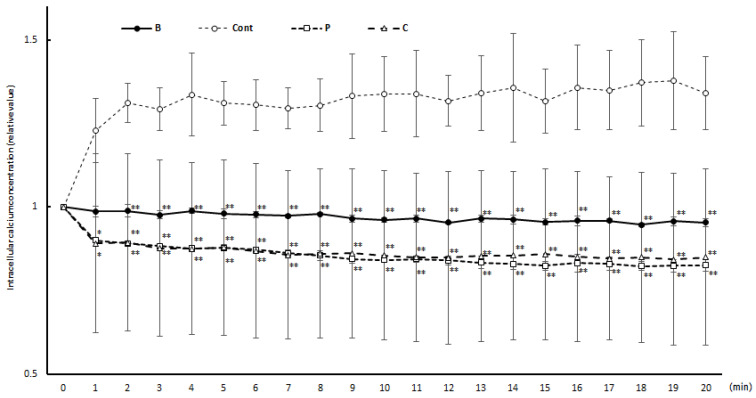
Intracellular calcium concentration over time in response to antigen–antibody reaction. In the degranul: ○Cont: control group, □O: perilla group, △C: chive group. Results are expressed as means ± standard deviation (*n* = 3). Data were analyzed using one-way ANOVA followed by Dunnett’s multiple comparisons; * *p* < 0.05 and ** *p* < 0.01 versus the Cont group.

**Figure 3 nutrients-16-00693-f003:**
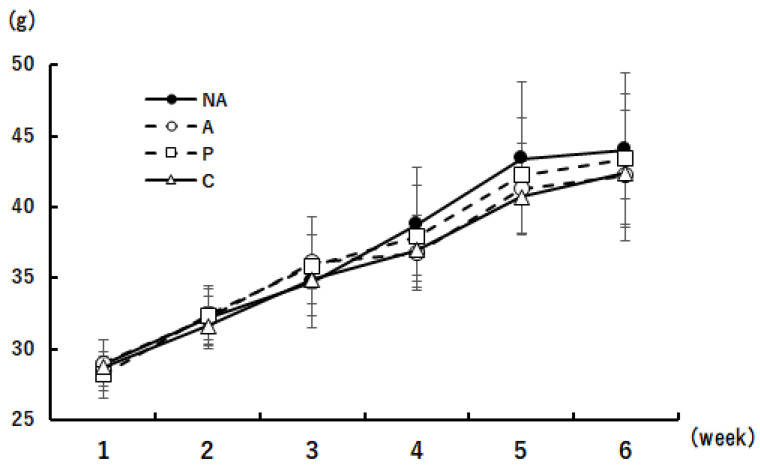
Body weight changes. groups k period iodple com: ○A: sensitized group, □O: perilla group, △C: chive group. Results are expressed as means ± standard deviation (NA group: *n* = 9, others: *n* = 8).

**Figure 4 nutrients-16-00693-f004:**
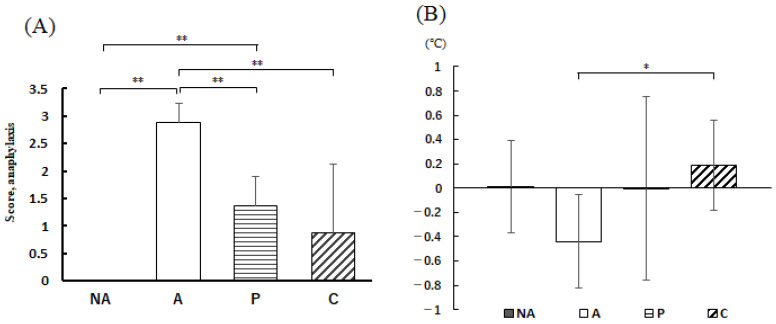
Anaphylaxis assessment. (**A**) Anaphylaxis score, (**B**) rectal temperature change 30 min after antigen administration, NA: non-sensitized group, A: sensitized group, P: perilla group, C: chive group. Results are expressed as means ± standard deviation (NA group: *n* = 9, others: *n* = 8). Data were analyzed using one-way ANOVA followed by Bonferroni multiple comparisons; * *p* < 0.05 and ** *p* < 0.01.

**Figure 5 nutrients-16-00693-f005:**
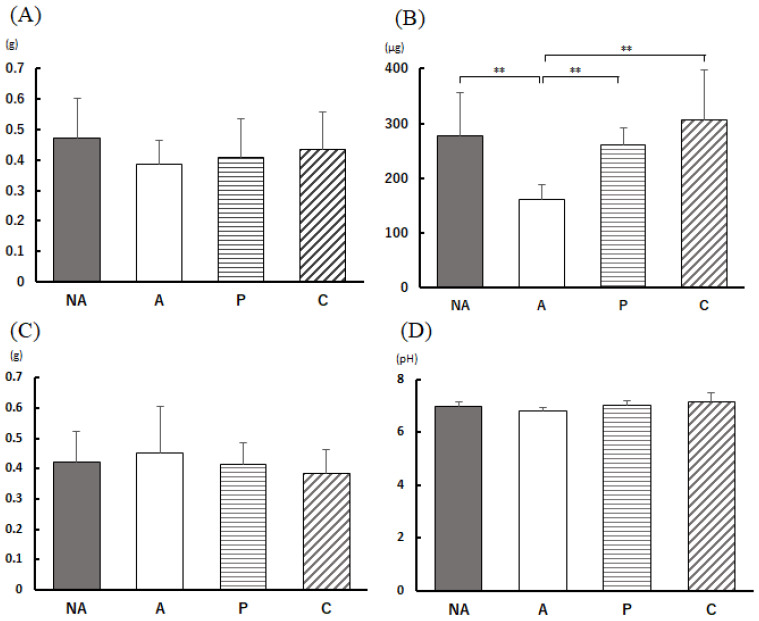
Feces and cecum parameters. (**A**) Fecal weight, (**B**) fecal IgA content, (**C**) cecum weight, (**D**) cecum pH, NA: non-sensitized group, A: sensitized group, P: perilla group, C: chive group. Results are expressed as means ± standard deviation (NA group: *n* = 9, others: *n* = 8). Data were analyzed using one-way ANOVA followed by Bonferroni multiple comparisons; ** *p* < 0.01.

**Figure 6 nutrients-16-00693-f006:**
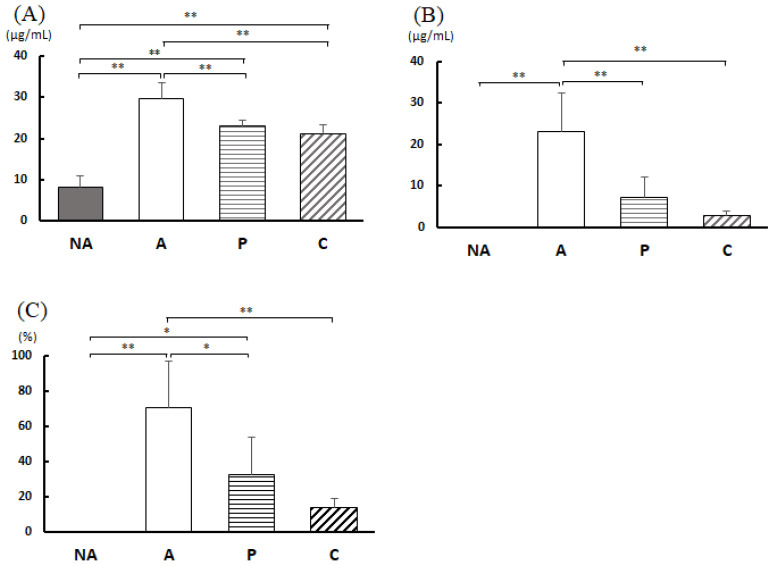
Blood antibody titer. (**A**) Total IgE, (**B**) anti-OVA IgE, (**C**) anti-OVA IgE/total IgE, NA: non-sensitized group, A: sensitized group, P: perilla group, C: chive group. Results are expressed as means ± standard deviation (NA group: *n* = 9, others: *n* = 8). Data were analyzed using one-way ANOVA followed by Bonferroni multiple comparisons; * *p* < 0.05 and ** *p* < 0.01.

**Figure 7 nutrients-16-00693-f007:**
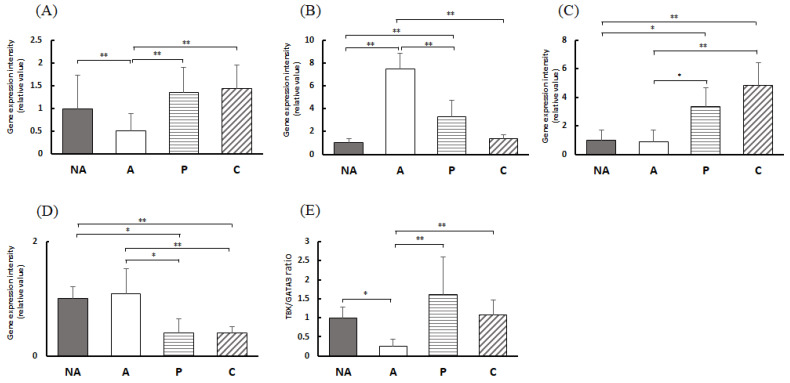
Colon T-cell transcription factor gene expression levels: (**A**) TBX1, (**B**) GATA3, (**C**) Foxp3, (**D**) Rorc, (**E**) TBX1/GATA3 expression ratio, NA: non-sensitized group, A: sensitized group, P: perilla group, C: chive group. Results are expressed as means ± standard deviation (NA group: *n* = 9, others: *n* = 8). Data were analyzed using one-way ANOVA followed by Bonferroni multiple comparisons; * *p* < 0.05 and ** *p* < 0.01.

**Figure 8 nutrients-16-00693-f008:**
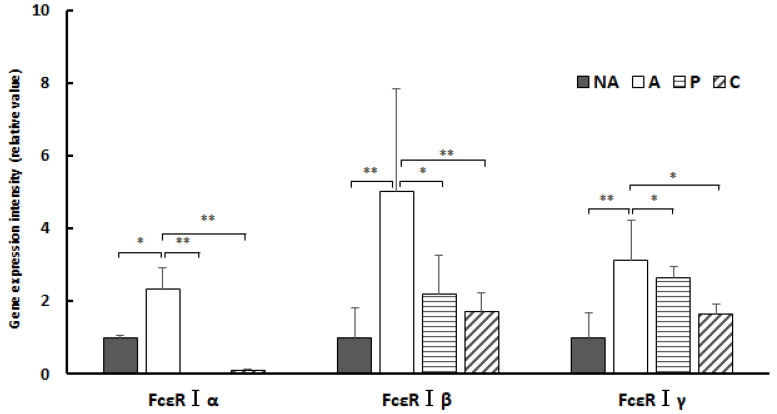
Colon IgE receptor gene expression: NA: non-sensitized group, A: sensitized group, P: perilla group, C: chive group. Results are expressed as means ± standard deviation (NA group: *n* = 9, others: *n* = 8). Data were analyzed using one-way ANOVA followed by Bonferroni multiple comparisons; * *p* < 0.05 and ** *p* < 0.01.

**Figure 9 nutrients-16-00693-f009:**
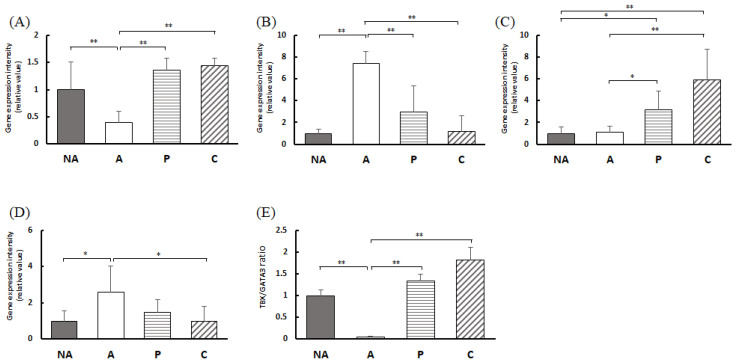
Splenic T-cell transcription factor gene expression levels: (**A**) TBX1, (**B**) GATA3, (**C**) Foxp3, (**D**) Rorc, (**E**) TBX1/GATA3 expression ratio, NA: non-sensitized group, A: sensitized group, P: perilla group, C: chive group. Results are expressed as means ± standard deviation (NA group: *n* = 9, others: *n* = 8). Data were analyzed using one-way ANOVA followed by Bonferroni multiple comparisons; * *p* < 0.05 and ** *p* < 0.01.

**Figure 10 nutrients-16-00693-f010:**
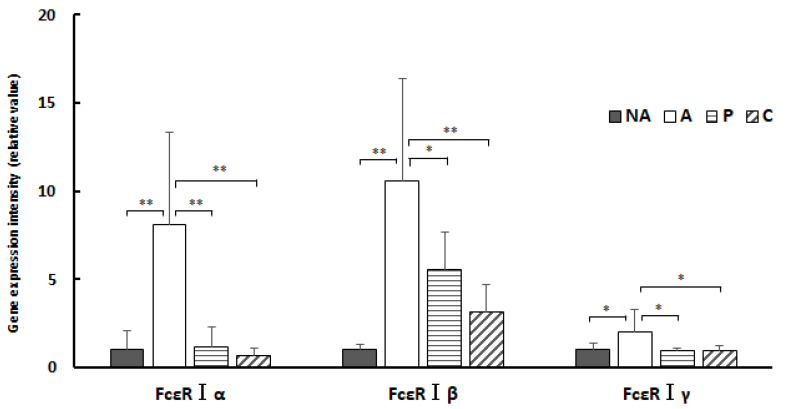
Splenic IgE receptor gene expression: NA: non-sensitized group, A: sensitized group, O: oba group, N: chive group. Values are mean ± standard deviation (NA group: *n* = 9, others: *n* = 8). A Bonferroni test was used to test statistically significant differences between groups *: * *p* < 0.05 and ** *p* < 0.01.

**Table 1 nutrients-16-00693-t001:** Feed mix ratio.

Feed Mixing List	AIN93G	Perilla Diet	Chive Diet
Milk casein	20.00	20.00	20.00
L-Cystine	0.30	0.30	0.30
Cornstarch	63.1986	61.1986	61.1986
Vegetable extract	0.0000	2.0000	2.0000
Edible oil	7.00	7.00	7.00
Cellulose powder	5.00	5.00	5.00
AIG-93 mineral mix	3.50	3.50	3.50
Contains AIG-93 vitamins (choline deuterate)	1.00	1.00	1.00
Tertiary butylhydroquinone	0.00140	0.00140	0.00140
Total	100.000	100.000	100.000

Values are expressed in (%).

**Table 2 nutrients-16-00693-t002:** Systemic anaphylactic symptom scores.

Score	Symptoms
0	No symptoms
1	Scratching and rubbing around the nose and head
2	Puffiness around the eyes and mouth, pilar erecti, reduced activity, and/or decreased activity with increased respiratory rate
3	Wheezing, labored respiration, and cyanosis around the mouth and the tail
4	No activity after prodding or tremor and convulsion
5	Death

**Table 3 nutrients-16-00693-t003:** Total polyphenol content and antioxidant capacity of 80% ethanol extracts of 22 vegetables.

Vegetable (Scientific Name)	Total Polyphenol Content (mg/g Chlorogenic Acid Equivalent)	Antioxidant Capacity (Mg/G L-Ascorbic Acid Equivalent)
Tomato (*Solanum lycopersicum*)	4.35 ± 0.11	1.13 ± 0.02
Bell pepper (*Capsicum annuum* ‘Grossum’)	17.97 ± 5.79	1.41 ± 0.00
Green pepper (*Capsicum annuum* ‘Grossum’)	19.14 ± 0.66	1.45 ± 0.00
Pumpkin (*Cucurbita maxima*)	4.11 ± 0.05	1.06 ± 0.01
Eggplant (*Solanum melongena*)	15.10 ± 1.10	1.43 ± 0.01
Bitter melon (*Momordica charantia*)	20.85 ± 0.30	1.45 ± 0.00
Snap pea (*Pisum sativum*)	5.00 ± 0.14	1.24 ± 0.01
Broccoli (*Brassica oleracea* var. *italic*)	22.69 ± 0.28	1.41 ± 0.00
Asparagus (*Asparagus officinalis*)	18.56 ± 0.38	1.41 ± 0.00
Welsh onion (*Allium fistulosum*)	5.22 ± 0.06	1.12 ± 0.01
Komatsuna (*Brassica rapa* var. *perviridis*)	32.52 ± 0.24	1.41 ± 0.00
Spinach (*Spinacia oleracea*)	17.56 ± 0.18	1.41 ± 0.00
Garland chrysanthemum (*Glebionis coronaria*)	32.38 ± 1.46	1.39 ± 0.00
Cabbage (*Brassica oleracea* var. *capitate*)	12.84 ± 0.14	1.45 ± 0.00
Perilla (*Perilla frutescens* var. *crispa* f. *viridis*)	115.94 ± 7.84	1.39 ± 0.00
Chives (*Allium tuberosum*)	9.51 ± 0.17	1.40 ± 0.01
Carrot (*Daucus carota* subsp. *sativus*)	3.87 ± 0.07	0.77 ± 0.03
Onion (*Allium cepa*)	4.55 ± 0.11	0.76 ± 0.02
Garlic (*Allium sativum*)	3.27 ± 0.01	0.63 ± 0.01
Myoga (*Zingiber mioga*)	5.62 ± 0.27	1.92 ± 0.01
Daikon (*Raphanus sativus* var. *hortensis*)	5.61 ± 0.08	1.45 ± 0.00
Ginger (*Zingiber officinale*)	12.63 ± 0.87	1.44 ± 0.00

Values are mean ± standard deviation (*n* = 3).

**Table 4 nutrients-16-00693-t004:** Cell-adapted concentrations and maximum β-hexosaminidase inhibition rates of 80% ethanol extracts of 22 vegetables.

Vegetable (Scientific Name)	Cell Adaptation Concentration (µg/mL)	Maximum Inhibition (%) (Concentration of Vegetable Extracts Added) ^1^
Tomato (*Solanum lycopersicum*)	≤250	0
Bell pepper (*Capsicum annuum* ‘Grossum’)	≤250	0
Green pepper (*Capsicum annuum* ‘Grossum’)	≤63	0
Pumpkin (*Cucurbita maxima*)	≤250	0
Eggplant (*Solanum melongena*)	≤250	0
Bitter melon (*Momordica charantia*)	≤16	0
Snap pea (*Pisum sativum*)	≤16	0
Broccoli (*Brassica oleracea* var. *italic*)	≤250	0
Asparagus (*Asparagus officinalis*)	≤250	0
Welsh onion (*Allium fistulosum*)	≤31	0
Komatsuna (*Brassica rapa* var. *perviridis*)	≤63	0
Spinach (*Spinacia oleracea*)	≤250	0
Garland chrysanthemum (*Glebionis coronaria*)	≤16	27 ± 4 (8 µg/mL)
Cabbage (*Brassica oleracea* var. *capitate*)	≤63	0
Perilla (*Perilla frutescens* var. *crispa* f. *viridis*)	≤500	44 ± 4 (500 µg/mL)
Chives (*Allium tuberosum*)	≤250	49 ± 4 (250 µg/mL)
Carrot (*Daucus carota* subsp. *sativus*)	≤31	0
Onion (*Allium cepa*)	≤63	0
Garlic (*Allium sativum*)	≤16	0
Myoga (*Zingiber mioga*)	≤63	19 ± 3 (8 µg/mL)
Daikon (*Raphanus sativus* var. *hortensis*)	≤125	0
Ginger (*Zingiber officinale*)	≤250	0

^1^ Values are mean ± standard deviation (*n* = 3).

## Data Availability

Data are contained within the article.

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
