# Peer review of "Vegetable Extracts as Therapeutic Agents: A Comprehensive Exploration of Anti-Allergic Effects"

_nutrients, 2024, doi:10.3390/nu16050693_

Round 1

Reviewer 1 Report

Comments and Suggestions for Authors

The paper by Takemoto  et al, entitled “Vegetable Extracts as Therapeutic Agents: A Comprehensive Exploration of Anti-Allergic Effects” outlines  new interesting activity exerted by vegetable extracts on allergy processes. Authors claim that the inhibition of degranulation following antigen FcεR cross-linking is decreased by the addition of  perrilla and chives extracts. Moreover allergy reactivity was measured in mice sensitized to OVA by intraperitoneal administration and previously supplemented with perrilla or chives extracts and a significant reduction in allergic manifestations was recorded in mice that had received the extracts. The Authors suggest that the effect is mainly due to polyphenols, citing results reported in other studies. On the other hand they observe that the anti-allergic activity was not related to the total polyphenol content or antioxidant capacity. As far as this issue is concerned, I  wonder  if the Authors   have assayed the effects of purified chlorogenic acid or resveratrol which have been studied extensively for their pleiotropic activities in different cellular models. As a second observation I wonder if Authors did add perrilla and chives together to ascertain if the effect would be synergic or simply additive.

As a minor point, I do not understand the meaning of the following sentence (line 504): “The results of this study suggest that in the spleen, as in the colon, Th cell transcription factors corrected the Th1/Th2 balance disrupted by antigen sensitization and elevated the expression of Th17 cells, which contribute to allergy induction”. Th17 cells promote inflammation and autoimmunity: did the extracts increase their numbers?

Reviewer 2 Report

Comments and Suggestions for Authors

The study deals with a significant issue. Methods are sound, and results clearly presented. 

Comment

Lines 440-441

It is highly hypothetical that suppression of temperature lowers the risk of an anaphylactic reaction. Pls provide respective data or omit
